# Immunotherapy of Hematological Malignancies of Human B-Cell Origin with CD19 CAR T Lymphocytes

**DOI:** 10.3390/cells13080662

**Published:** 2024-04-09

**Authors:** Darya Khvorost, Brittany Kendall, Ali R. Jazirehi

**Affiliations:** 1Department of Life Sciences, Los Angeles City College (LACC), 855 N. Vermont Ave., Los Angeles, CA 90029, USA or kendalbd3388@laccd.edu (B.K.); 2Department of Biological Sciences, College of Natural and Social Sciences, California State University, Los Angeles (CSULA), Los Angeles, CA 90032, USA

**Keywords:** immunotherapy, apoptosis, signal transduction, chromatin remodeling, Acute Lymphoblastic Leukemia, non-Hodgkin’s lymphoma, celecoxib

## Abstract

Acute lymphoblastic leukemia (ALL) and non-Hodgkin’s lymphoma (NHL) are hematological malignancies with high incidence rates that respond relatively well to conventional therapies. However, a major issue is the clinical emergence of patients with relapsed or refractory (r/r) NHL or ALL. In such circumstances, opportunities for complete remission significantly decline and mortality rates increase. The recent FDA approval of multiple cell-based therapies, Kymriah (tisagenlecleucel), Yescarta (axicabtagene ciloleucel), Tecartus (Brexucabtagene autoleucel KTE-X19), and Breyanzi (Lisocabtagene Maraleucel), has provided hope for those with r/r NHL and ALL. These new cell-based immunotherapies use genetically engineered chimeric antigen receptor (CAR) T-cells, whose success can be attributed to CAR’s high specificity in recognizing B-cell-specific CD19 surface markers present on various B-cell malignancies and the subsequent initiation of anti-tumor activity. The efficacy of these treatments has led to promising results in many clinical trials, but relapses and adverse reactions such as cytokine release syndrome (CRS) and neurotoxicity (NT) remain pervasive, leaving areas for improvement in current and subsequent trials. In this review, we highlight the current information on traditional treatments of NHL and ALL, the design and manufacturing of various generations of CAR T-cells, the FDA approval of Kymriah, Yescarta Tecartus, and Breyanzi, and a summary of prominent clinical trials and the notable disadvantages of treatments. We further discuss approaches to potentially enhance CAR T-cell therapy for these malignancies, such as the inclusion of a suicide gene and use of FDA-approved drugs.

## 1. Acute Lymphoblastic Lymphoma

ALL is a malignancy of hematopoietic stem cells and accounts for 30% of all childhood cancers, making it the most common cancer occurring in children. ALL presents when there is an accumulation of malignant and immature lymphoid cells, primarily in the bone marrow and systemically in the blood [1]. ALL is not just limited to children and young adults; however, it does have a peak incidence in those aged 2–5. ALL accounts for 15% of adult leukemia incidences and peaks after 50 years old. The prognosis for recovery in children is high, with overall survival rates of around 90%, but infants and adults exhibit poorer outcomes [1,2]. ALL can occur in T lymphoid precursors (15% of childhood ALL and 25% of adult ALL) or B cell lymphoid precursors and is mainly attributed to chromosomal abnormalities such as translocations or aneuploidy. Germline abnormalities are also found in both the neoplasms and healthy cells of patients. Environmental risk factors such as exposure to some chemicals or radiation have been linked to ALL, but in very few cases [2,3,4]. Common symptoms include bleeding or bruising due to thrombocytopenia, infection from neutropenia and pallor/fatigue due to anemia [2]. 

ALL mainly targets the hematopoietic stem cells but it can spread to organs like the spleen and liver, as well as the CNS and lymph nodes [5]. The classification of ALL subtypes is determined by two different systems: the World Health Organization (WHO) and the French American and British (FAB) system. WHO determines the ALL subtype according to its precursor cell while the FAB system classifies ALL as L_1_, L_2_, or L_3_ according to its morphology and histology [6]. L_1_’s morphology comprises small cells, sparse cytoplasm, a uniform chromatin, a nuclear shape (some indentation could be present), and very small nucleoli. L_2_′s morphology comprises overall larger cells, varying in size, chromatin distribution, a large quantity of cytoplasm, and a nuclear shape with one or more pronounced nucleoli. L_3_′s cytological features show large and uniformly shaped cells with homogenous and somewhat stippled chromatin, a round or oval shape, and relatively ample cytoplasm. L_1_ and L_2_ can be of either B or T cell linages, while L_3_ is rare, with only 1–2% of ALL cells exhibiting this morphology, which is mainly associated with mature B-lymphocytes. A more modern approach is through the WHO, which categorizes ALL into B-cell or T-cell linages and considers genetics, clinical manifestations, and immunophenotyping in addition to cytomorphology [7,8]. 

## 2. Standard Treatment Options for Acute Lymphoblastic Leukemia (ALL)

Treatment plans for acute lymphoblastic leukemia (ALL) have evolved as long-term side effects have been defined and new treatment options have been discovered.

ALL patients at high risk of central nervous system (CNS) relapse were typically treated with cranial radiation therapy (CRT) followed by intensive chemotherapy. This treatment plan historically resulted in an up to 80% 5-year-event-free survival rate in standard risk ALL patients [9]. During radiation therapy, intense bundles of energy in the form of an X-ray beam are fired at cancer cells to damage the cell DNA and cause the cells to die. Chemotherapy employs a series of drugs that inhibit the cell division of rapidly dividing cells, leading to their death. A chemotherapy and irradiation combination treatment plan is relatively successful at eradicating cancer, but neither treatment is specific to cancer cells. Therefore, both treatments damage healthy cells as well as cancerous cells, and cause unwanted side effects. CRT specifically has many serious later effects, including neurocognitive deficits, endocrinopathies, and secondary brain tumors, which occur in more than two-thirds of long-term survivors [10,11]. 

Following the discovery of the late effects of CRT, standard ALL treatment shifted to intravenous, intrathecal, and systemic chemotherapeutic drug administration, omitting prophylactic cranial irradiation. A clinical trial following the proposal of this new treatment plan consisted of 498 patients who were treated with chemotherapy alone, which resulted in an 85.6% 5-year-event-free probability and 93.5% overall survival probability [11]. Typical chemotherapy treatment protocols are divided into two four-week cycles. The first phase involves cyclophosphamide on the first day, 3 consecutive days of daunorubicin, biweekly L-asparaginase, weekly vincristine, and 3 weeks of prednisone. The second phase introduces cyclophosphamide with cytarabine, oral 6-mercaptopurine (6-MP), four doses of methotrexate, and cranial radiation [4]. Following the induction phases, patients undergo 3–4 cycles of intensification therapy with methotrexate. Although this contemporary chemotherapeutic trial achieved unprecedented survival probabilities, patients receiving this treatment still experienced a variety of treatment-related side effects, including osteonecrosis and cardiovascular and endocrine morbidity, as well as neurocognitive deficits [9]. 

ALL patients treated with chemotherapy often develop resistances and require chemo-sensitizing agents for the treatment to remain effective. Patients with Ph+ ALL currently start with a treatment of chemo-sensitizing imatinib mesylate combined with chemotherapy. Some patients experience a relapse after their immune system develops resistance to this treatment protocol and require the addition of dasatinib, a second-generation BCR-ABL tyrosine kinase inhibitor, to effectively treat their cancer [12]. Dasatinib allows the drug to more efficiently penetrate the blood–brain barrier; however, in clinical trials, cases of BCR-ABL-mutated dasatinib-resistant clones were observed, resulting in the need for another second-generation BCR-ABL tyrosine kinase inhibitor, nilotinib, for chemotherapy to remain effective [12]. 

Several cycles of chemotherapy administration, followed by the selective outgrowth of chemotherapy-resistant clones, and finally a new, slightly altered chemotherapy treatment is detrimental to human health. As previously mentioned, chemotherapy targets rapidly dividing cells, not cancer cells exclusively; these include blood-forming cells in the bone marrow, hair follicles, and cells in the reproductive system, mouth, and digestive tract. Some of the effects of chemotherapy are short-lived and will cease upon completion of the treatment, such as increased susceptibility to infection and mucositis. However, some side effects of high-dosage chemotherapy can be long-lasting and even fatal. These include venoocclusive disease of the liver (VOD), interstitial pneumonia syndrome (IPS), and increased risk of new cancer development [13]. This highlights the need to develop treatment options that will have fewer side effects. 

Allogenic hematopoietic stem cell transplant (AHSCT) is another treatment option for ALL patients and is used to suppress the disease and restore the patient’s immune system. AHSCT must be preceded by chemotherapy or total-body irradiation (TBI) in ALL patients to eradicate as many cancer cells as possible and to provide a place for the healthy transplanted cells to grow and divide [14]. The typical dosages of TBI range from 13.5 Gy to 8 Gy, with the most common dosage being 12 Gy [15]. A recent clinical trial recorded a 65% 8-year overall survival in ALL patients of 30 years old or younger who were treated with this protocol [13]. Allogeneic stem cell transplantation reduces relapse rates in ALL patients, but transplantation includes toxic side effects. These side effects include all of those from high-dosage chemotherapy as well as graft-versus-host disease, where donor lymphocytes attack the recipient’s cells, graft rejection ensues, and bone marrow function is not reinstated. Additionally, women treated with TBI before the transplant have an extremely high risk of becoming sterile and developing cataracts [13]. The severity of the side effects is directly correlated to the degree of donor/recipient HLA-matching, the age of the patient, the type of high-dosage chemotherapy administered prior to treatment, and the degree of the patient’s immune system suppression. 

## 3. Non-Hodgkin’s Lymphoma (NHL)

Non-Hodgkin’s Lymphoma (NHL) is a cancer of precursor lymphocytes and results in solid tumors of the lymphoid tissue at certain lymphocyte differentiation stages. It has no conclusive etiology [16]. More than half of all cases occur in adults over 65 years of age, but it is also common amongst adolescents (ages 15–19) and young adults (ages 20–39), accounting for 7% and 6% of cancer incidences, respectively [17]. Collectively, NHL makes up 4.3% of all cancers in the United States, making it the most frequently occurring hematologic malignant neoplasm in the country. It is estimated that, in 2020, there will be 77,240 new cases and 19,940 deaths. Mortality rates were the highest in 1997 but by 2004 they had dropped from 11.1 to 7.0 per 100,000 population, respectively. Lymphoma rates in general dropped by 80% in children and 82% in adolescents from 1970 to 2017 [18,19,20,21]. The risk factors are not yet well-understood, but studies suggest that infection and poor immune systems are the most probable causes of the disease. High NHL rates have been observed in people with some autoimmune diseases as well as diseases that deplete the immune system, such as HIV/AIDS. Rarer NHL subtypes like Epstein–Barr virus and Burkitt lymphoma have been associated with the incidence of infection. Environmental and lifestyle factors have been suggested to increase the risk of NHL, depending on the subtype, but some of these results are not conclusive [22]. NHL is categorized as either low-grade or high-grade, with each containing specific subtypes of NHL. Low-grade or indolent NHL includes follicular lymphoma, mantel cell lymphoma, marginal cell lymphoma, small lymphocytic lymphoma, lymphoplasmacytic lymphoma, enteropathy-associated T cell lymphoma, and skin lymphoma. High-grade NHL consists of diffuse large B cell lymphoma (DLBCL), Burkitt lymphoma, mediastinal large B cell lymphoma, and peripheral T cell lymphoma. Rarer types of high- and low-grade NHL also exist [23,24,25]. While both are serious subtypes of this cancer, the difference between low-grade and high-grade NHL is the initial aggressiveness. Low-grade NHL spreads slowly compared to high-grade NHL, and differences also exist in the therapies established for treatment. Low-grade NHL also has the potential to become high-grade, making its treatment approach more aggressive. This transformed NHL has a poorer prognosis. Most NHL types (85–90%) are B cell NHL and the remaining 10–15% are T cell NHL. The majority of patients initially exhibit B type symptoms, such as night sweats, bone pain, fatigue, and fever. Lymphadenopathy (occurrence may fluctuate) and the extranodal appearance of lumps elsewhere on the body, commonly on the skin or in the GI tract, are also common. Symptoms vary by subtype and disease advancement [24,25,26,27]. The International Prognosis Index is used in combination with the Ann Arbor Staging System to determine clinical risk factors when evaluating a newly diagnosed patient. The model looks at how many extranodal sites are present, serum lactate dehydrogenase concentration, the age of the patient (either over 60 years old or younger than 60 years old), how advanced the tumors are according to the Ann Arbor system, and the patient’s Eastern Cooperative Performance Status. The World Health Organization classification of NHL subtype is the most commonly used system in the world and is implemented through the excisional biopsies of affected lymph nodes or other affected lymphatic tissues [28].

## 4. Traditional Treatments for Non-Hodgkin’s Lymphoma (NHL)

According to the American Cancer Society, the 5-year survival percentage of NHL patients from 2008 to 2014, from birth to 14 years old, is 90.2%, and slightly decreases to 89.1% from 14 to 19 years of age [21]. About 90–100% of ALL cases in children attain complete response (CR), and over 80% of pediatric patients with a new ALL diagnosis (following conventional regimens) are cured. However, adults with ALL have a cure rate of only 50–60%, even though their CR is 90% [29]. One of the two traditional treatments for NHL is a chemotherapy regimen, cyclophosphamide, doxorubicin, vincristine, and prednisone (CHOP), which has shown an OS of 35% and a CR rate in elderly patients of 40% [30]. A study of 225 patients with intermediate- to high-grade NHL, ranging in age from 15 to 71 (median 56), compared the efficacy of CHOP and three other intensive chemotherapies (total *n* = 899, with each treatment group containing at least *n* = 218). At three years, the disease-free OS was estimated at 54% for the CHOP patients, with no significant increase in OS in comparison to the other treatments in the study. CHOP did, however, show lower levels of toxicity, and was therefore still considered the standard of care at the time of publication (April 1993) [31]. Another regimen, R-CHOP (CHOP plus rituximab), has shown important improvements in efficacy for the treatment of NHL. Rituximab (Rituxan), a CD20 monoclonal antibody, was discovered in 1991 but not approved for use in the treatment of NHL by the FDA until 1997 [32]. An evaluation of 824 patients in two groups of 18- to 60-year-olds revealed that the 3-year event-free-survival rate of those in the group treated with R-CHOP vs. CHOP alone was 20% higher (79% and 59%, respectively) [33]. A 13% higher CR was observed in elderly patients treated with R-CHOP vs. CHOP alone [34]. 

The conventional standard-of-care chemotherapy treatment for ALL patients is generally conducted in several phases depending on their relapse risk assessment: induction chemotherapy (4–6 weeks), combination chemotherapy for numerous months, and maintenance chemotherapy lasting 18–30 months are standard for lower-risk patients. Interim maintenance and delayed intensification phases are added to the protocol for patients with the highest risk of relapse [2,29]. For remission induction, prednisolone is the most commonly used glucocorticoid, followed by dexamethasone for the reintensification phase [35]. Despite the promising outcomes of R-CHOP for the treatment of NHL, and the promising outcomes of combination chemotherapy for ALL patients, a significant decrease in cure rates occurs for those who develop resistance or relapse after or during treatment [2,36]. Collectively, these results highlight the need for the design of novel approaches, with more efficacy and less toxicity), for the treatment of ALL and NHL (Figure 1). 

## 5. Design of Various CD19 CAR Constructs

The specificity of the CD19 antigen has helped chimeric antigen receptor (CAR) T-cells succeed in early clinical trials. Apart from healthy B-cells, normal tissues and hematopoietic stem cells do not exhibit the CD19 antigen [37]. This allows the CD19 antigen to be properly directed towards targeting the B-cell leukemias and lymphomas that do express it. Antibodies and T-cells are hybridized and genetically engineered into CAR T-cells that consist of an extracellular single-chain variable fragment (scFv), a cell-membrane-spanning region, and cytoplasmic domains [38]. The extracellular antigen-specific scFv has V_H_ and V_L_ genes, which are attached by a disulfide bond or peptide linker [39]. CD8 or IgG4 molecules make up the intracellular portion of the cell, along with a CD3ζ-signaling domain in the cytoplasm region [38]. The previously described backbone of CAR-T cells has remained consistent from inception in the first-generation versions through the later fourth-generation versions. Tisagenlecleucel and axicabtagene ciloleucel are second-generation (2G) CAR T-cell immunotherapies that have both a CD3ζ signaling domain and one intercellular costimulatory region. The coupling of both cytoplasmic regions is responsible for actuating the T-cell. In axicabtagene ciloleucel, the addition of the signal transduction domain, CD28, in the cytoplasm mimics the CD28/B7 interactions shared by tumor cells and T lymphocytes [40]. With this, the CARs have T-cells with heightened cytotoxicity for tumor-specific antigens [41]. In tisagenlecleucel, the costimulatory domain in the cytoplasmic region of the CAR T-cell is 4-1BB. Similarly, the addition of this costimulatory region intensifies cytokine production and propagation while increasing the persistence of the CAR T-cells in the body [42]. Another 2G CAR T-cell was engineered for r/r B-ALL and has a 1:1 ratio of CD8^+^- and CD4^+^-specific subsets along with the 4-1BB and CD3ζ domains [43]. Third-generation (3G) CAR T-cells integrate both CD28 and 4-1BB (CD137) or CD28 and OX40 (CD134) as costimulatory domains [44]. It is not yet known if the unification of CD28 and 4-1BB helps the CAR T-cells to expand and work more efficiently than their 2G versions. However, since, individually, the 2G costimulatory domains trigger different pathways in T-cells, which have shown success in treating r/r ALL or r/r NHL, the idea is promising. Anti-tumor activity was shown to be enhanced when combining CD28 and 4-1BB, and the results from a recent clinical trial comparing 2G and 3G CAR T-cell activity together in vivo showed a beneficial outcome for patients with r/r NHL [45,46,47]. Fourth-generation CARs, called TRUCKs, are T-cells that have been designed to have an inducible transgenic product that they transport to the targeted tumor. Apart from having the scFv, CD3ζ, and a costimulatory region, they express a transgenic cytokine such as IL-12. This is achieved by an activated T-cell’s nuclear factor (NFAT). The cytokine gene cassette can then excrete cytokines (IL-12 or others in the same family), which strengthen immune responses and equip the CAR T-cell with superior anti-tumor mechanisms [48,49]. Fifth-generation CAR-T cells have membrane receptors. Although various approaches have been tested, the most promising one is the addition of the IL-2 receptors, allowing, in an antigen-dependent fashion, for the activation of the JAK/STAT signal transduction [50] (Figure 2).

## 6. Tisagenlecleucel and Axicabtagene Ciloleucel for the Treatment of Refractory Non-Hodgkin’s Lymphoma and Acute Lymphoblastic Leukemia

The years 2017 and 2018 marked a great advance for anti-CD19 chimeric antigen receptor (CAR) T-cell therapy, with the US Food and Drug Administration approving two drugs: Tisagenlecleucel (Kymriah; Novartis, East Hanover, NJ, USA), approved in August 2017, and axicabtagene ciloleucel (Yescarta; Kite Pharma, El Segunda, CA, USA), approved in May 2018 [50]. Both cell-based immunotherapies were developed for patients with relapsed or refractory (r/r) aggressive B-cell malignancies who had at least two unsuccessful prior lines of systemic therapy [50]. A third CAR T-cell drug, lisocabtagene maraleucel (liso-cel, JCAR-017), is in the final stages of clinical development [51]. Specifically, tisagenlecleucel is approved to treat patients under 25 years of age with pediatric or young adult B-cell acute lymphoblastic leukemia (ALL), while axicabtagene ciloleucel is approved for the treatment of r/r primary mediastinal large B-cell lymphoma (PMBCL). Both axicabtagene ciloleucel and tisagenlecleucel obtained approval extending to adult patients with a variety of non-Hodgkin’s Lymphomas (NHL), including diffuse large B-cell lymphoma (DLBCL) not otherwise specified, DLBCL arising from follicular lymphoma (tFL), and high-grade B-cell lymphoma [52,53,54].

## 7. Summary of Clinical Trials

Several recent and major clinical trials, such as ELIANA, JULIET, and ZUMA I/II, were conducted to test the efficacy of tisagenlecleucel and axicabtagene ciloleucel in NHL and ALL treatments. These clinical trials are summarized in Table 1 and some of the prominent ones are discussed in Table 1 and Table 2 [51,52,53,54,55,56,57,58,59,60].

## 8. CD19 CAR-T Therapy in NHL

CAR T-cell therapy has had a recent exciting breakthrough, with two CAR T-cell products receiving FDA approval: axicabtagene ciloleucel (Yescarta) for DLBCL and tisagenlecleucel (Kymriah) for adult DLBCL subtypes and pediatric ALL. Clinical trials using these two agents have shown remarkable efficacy while increasing the homogeneity of treatment, resulting in more predictable toxicities and reliable post-administration proliferation (Table 2).

In a trial conducted at the University of Pennsylvania, investigators treated a total of 14 patients with DLBCL and 14 patients with follicular lymphoma (FL) using tisagenlecleucel [56]. A high percent of both groups had refractory disease: 86% of DLBCL and 57% of FL patients. Complete response (CR) was seen in 6 of 14 DLBCL patients (43%) and 10 of 14 FL patients (71%). At a median follow-up of 28.6 months, 57% of all patients remained progression-free survival (95% CI, 36 to 73), including 70% of patients in the FL group and 43% of DLBCL patients. Cytokine release syndrome (CRS) and neurotoxicity (NT) remained the most common adverse events, although CRS of grade 3 or higher was low, at 18%. 

The ZUMA-1 trial by Kite evaluated axicabtagene ciloleucel in seven patients with refractory DLBCL who had no remaining chemotherapy treatment options and found CR in four patients (57%) and an overall response (OR) in five patients (71%), with three patients (43%) remaining disease-free at 12 months [54]. 

The ZUMA-2 trial funded by Kite Pharma and the Leukemia and Lymphoma Society Therapy Acceleration Program showed comparable results to JULIET. A total of 111 patients who had r/r DLBCL, tFL, or PMBCL with prior lines of therapy were considered for the ZUMA II study, and only 69% of patients had three or more lines of therapy. The treatment, axicabtagene ciloleucel, was administered to 101 patients (91%) with a median age of 58 years. After a conditioning chemotherapy treatment, a single target dose of 2 × 10^6^ CAR T-cells per kg of body weight was infused intravenously. As a result, the ORR at or above a 6-month follow-up was 82%, with 54% having a CR and 28% having a PR. The remainder had either stable disease, disease progression, or were unevaluable. At a median follow-up of 15.4 months, the ORR was 42% and the CR was 40%. By 18 months, the survival rate was determined to be 52%. Among several adverse effects, CRS of any grade affected 93% of patients (13% had grade 3 or higher) and 64% experienced neurologic events such as encephalopathy (64%, any grade). Three patients died while undergoing treatment (two from CRS; one from pulmonary embolism). An updated analysis showed that 37 patients died from disease progression and 4 died from other lines of therapy after axicabtagene ciloleucel in combination with disease progression [57,58]. 

## 9. CD19 CAR-T Therapy in ALL

In two trials of 16 and 30 patients with relapsed or r/r B-cell ALL (*n* = 16 comprised only adults with a median age of 50; *n* = 30 comprised 26 children/young adults and 4 adults > 22 year), both achieved CR, with rates ≥ 90% (51, 52). In the 30-patient study conducted at the Children’s Hospital of Philadelphia and the Hospital of the University of Pennsylvania using tisagenlecleucel, CRS was experienced by 100% of the patients, with 27% of cases being severe (grade 3 or higher); however, the investigators noted that the C-reactive protein (CRP) functioned as a reliable indicator for CRS severity and allowed for successful abatement with tocilizumab [58,59]. The OS rate was 78% at 6 months. Thirteen patients had NT of any grade that included delirium and encephalopathy. At the median follow-up, no deaths were attributed to tisagenlecleucel; however, 23% of patients died from disease progression or relapse [59,60]. 

A total of 53 adults with B-ALL received Kymriah at the Memorial Sloan Kettering Cancer Center [60]. CR was seen in 44 patients (83%). Median overall survival was 12.9 months, and patients with low disease activity (predicted by blasts of <5%) saw a median survival of 20.6 months. A predictor of adverse events such as CRS and NT included marrow blasts > 5% [61]. The ELIANA trial, funded by Novartis Pharmaceuticals, was a phase II international cohort that investigated the efficacy of CTL-019 in 75 children and young adults averaging at 11 years old. Relapsed/refractory (r/r) B-ALL found an overall survival at 6 months of 90% and a 12-month survival of 76% [64]. The best overall response rate (ORR) at 3 or more months was 81%, with a complete response (CR) of 60% and partial response (PR) of 21%. CRS occurred in 77% of patients: 46% had CRS of grade 3 or higher. Because of CRS, 48% of patients required the administration of tocilizumab [64]. A total of 40% of patients had neurologic events of all grades except grade 4, which had no occurrences. Nineteen patients died post infusion, with the highest occurrences of death being 30 days after the infusion, primarily from B-cell ALL progression or relapse. CTL-019 was shown to be effective in another recent trial with a similar patient cohort of 59 r/r B-ALL patients aged 1.5–24 years who received 4-1BB CTL-019 infusions as a single-agent treatment. A total of 55 of 59 patients (93%) experienced complete remission, with a 79% overall survival at 12 months [59]. The results of these trials are summarized in Table 2.

## 10. Clinical Trials of Tisagenlecleucel and Axicabtagene Ciloleucel for the Treatment of Refractory Non-Hodgkin’s Lymphoma and Acute Lymphoblastic Leukemia

The major clinical trials that were conducted to test the efficacy of tisagenlecleucel and axicabtagene ciloleucel are ELIANA, JULIET, and ZUMA I/II [55,56,57]. These trials are discussed below and summarized in Table 3.

In the ELIANA trial funded by Novartis Pharmaceuticals, 92 patients were enrolled and 75 were treated with tisagenlecleucel. The phase II international cohort consisting of pediatric and young adult patients averaging 11 years old showed that the OS at 6 months was 90% and OS at 12 months was 76%. The best overall response rate (ORR) at 3 or more months was 81%, with a complete response (CR) of 60% and partial response (PR) of 21%. A variety of grade 3 or 4 adverse events attributed to tisagenlecleucel occurred in 73% of patients. Cytokine release syndrome (CRS) was experienced by 77% of patients, with 46% being grade 3 or higher and, because of CRS, 47% of the patients were treated in the ICU for an average of 8 days with a combination of treatments including tocilizumab for 37%. A total of 40% had neurologic events of all grades except grade 4, which had no occurrences. These events were treated with supportive care. Nineteen patients died post infusion, with the highest occurrences of death being 30 days after the infusion, primarily from B-cell ALL progression or relapse [64]. 

JULIET, a phase II, international study (funded by Novartis Pharmaceuticals), evaluated 93 adult patients with r/r DLBCL who continued to have disease progression after at least two lines of therapy, including rituximab and anthracycline. This suggests that post rituximab and anthracycline, patients were among the 20–35% of patients with the disease that relapsed, were among the 10–15% of patients to have primary refractory disease, or were unqualified for autologous transplantation. There were two cohorts in this study: one was in the United States (main cohort) and the other within the European Union (cohort A). The treatment used was centrally manufactured tisagenlecleucel. Of the 238 patients that were screened, 165 were enrolled, 111 were infused with a median dose of 3.0 × 10^8^ CAR-positive viable T cells, and 93 received an infusion with evaluation for efficacy. A total of 79% of the patients had DLBCL and 19% had tFL. Together, they had a median age of 56 (ranging from 22 to 76). More than half of the patients (56%) had stage IV disease upon entering the study. The ORR, according to an independent review committee, was 52% at 14 months (median time) from infusion to data cutoff. A total of 40% of patients had a CR with a PR rate of 12%. Disease-free progression at 12 months was estimated to be 79% for patients who exhibited a CR and 65% for patients who had a PR. CRS of any grade was the most common adverse event, presenting in 58% of the patients, followed by infection, where only 20% were grade 3 or higher. After 30 days of infusion, three patients died, and investigators determined that their cause of death was unrelated to tisagenlecleucel [59]. 

The ZUMA studies funded by Kite Pharma and the Leukemia and Lymphoma Society Therapy Acceleration Program showed comparable results to JULIET. One hundred and eleven patients who had r/r DLBCL, tFL, or PMBCL with prior lines of therapy were considered for the study. a total of 69% of patients had three or more lines of therapy. The treatment, axicabtagene ciloleucel, was manufactured for 110 patients but administered to 101 (91%) with a median age of 58 years. After a conditioning chemotherapy treatment, a single target dose of 2 × 10^6^ CAR T-cells per kg of body weight was infused intravenously. As a result, the ORR at or above a 6-month follow-up was 82%, with 54% of patients having a CR, and 28% having a PR. The remainder had either stable disease, disease progression, or were unevaluable. By the median follow up of 15.4 months, the ORR was 42% and the CR was 40%. By 18 months, the survival rate was determined to be 52%. Among several adverse effects, CRS of any grade affected 93% of patients (13% had grade 3 or higher) and 64% had neurologic events such as encephalopathy (64%, any grade). Three patients died while undergoing treatment (2 from CRS; 1 from pulmonary embolism). An updated analysis showed that 37 patients died from disease progression and 4 died from other lines of therapy after axicabtagene ciloleucel in combination with disease progression [58].

An additional study analyzed 30 patients (*n* = 25 of pediatric ages and *n* = 5 ranging from 26 to 60 years of age) with r/r ALL in a pilot clinical trial. The trial was conducted at the Children’s Hospital of Philadelphia and the Hospital of the University of Pennsylvania using tisagenlecleucel. At a median follow-up of 7 months, 90% of patients had a CR. The OS rate was 78% at 6 months. CRS of any grade was experienced by 100% of patients, and CRS of grade 3 or higher was experienced by 27%. Severe CRS was treated with tocilizumab. Thirteen patients had NT of any grade that included delirium and encephalopathy. At the median follow-up, no deaths were attributed to tisagenlecleucel; however, 23% of patients had died from disease progression or relapse [64]. The results of these trials are summarized in Table 3.

## 11. Summary of Clinical Trials of Brexucabtagene Autoleucel (Tecartus) for the Treatment of ALL and NHL

The clinical trials of brexucabtagene autoleucel (Tecartus) for the treatment of ALL and NHL are summarized in Table 4 [62,63,66,67,68]. The ZUMA-3 clinical trial (phase I/II) evaluated brexucabtagene autoleucel (Tecartus) in adult patients with r/r precursor B-cell ALL. The approval of Tecartus by the FDA was based on the phase II portion of the trial. The efficacy analysis population consisted of 54 patients who were enrolled in the phase II ZUMA trial (*n* = 54 comprised adult people within the range of 19–84 years, with the median age of 40 years, a slight predominance of men over women, and a predominance of Caucasians). The efficacy of the treatment was established based on complete remission (CR) being reached within 3 months after the infusion. A total of 52% of the 54 patients achieved CR within three months after infusion. The median time to CR was 56 days, with a range of 25–86 days. The safety population consisted of 78 people that were enrolled and treated in ZUMA-3 phase I or II trials. Among them, serious adverse reactions were observed in 79% and fatal adverse reactions happened in 5%. CRS of any grade occurred in 92%, while CRS of grade ≥ 3 was observed in 26%. Neurological toxicities (NT) of any grade occurred in 87%, and NT of grade ≥3 occurred in 35%. The median OS was 25.4 months [66].

The ZUMA-3 clinical trial (phase II) evaluated the efficacy and safety of brexucabtagene autoleucel (Tecartus) in adult patients with heavily pretreated r/r mantle cell lymphoma, with a median follow-up of 35.6 months. Of the 74 patients enrolled in the trial, only 68 received Tecartus (*n* = 68). The ORR was 91%. CR was 68% and PR was 24%. The median OS among patients was 46.6 months. The trial revealed that 99% of patients experienced toxicities. CRS of grades 1 or 2 occurred in 76% and CRS of grade ≥ 3 occurred in 15%. NT of grades of 1 or 2 occurred in 32% and NT of grade ≥ 3 was observed in 31%. Other toxicities included cytopenia (94%) and infections (32%). No deaths resulted from CRS or NT. The total number of deaths was 16 (24%). Death from progressive disease occurred in 14 patients (21%) [63,67]. The results of these trials are summarized in Table 4.

## 12. Summary of Clinical Trials of Lisocabtagene Maraleucel (Breyanzi) for the Treatment of NHL

The global TRASFORM phase III trial evaluated the efficacy and safety of lisocabtagene maraleucel (Breyanzi) in adult patients with r/r large B-cell lymphoma (LBCL) ≤ 12 months after first-line therapy intended for autologous stem-cell transplantation (ASCT) and compared it with the standard of care (SOC). The patients enrolled in the TRANSFORM phase 3 trial (*n* = 92) had a median age of 60 years with range of 20–74 years. In this study, with a 17.5-month median follow-up, the efficacy of treatment was established based on an ORR of 87% and a CR rate of 74%, and the median overall survival was not reached. The safety of the drug was confirmed by the fact that only Grade 3 CRS and NT occurred in 1% and 4% of patients, respectively. There were no grade 4 or 5 events. Prolonged cytopenia was experienced by 43% of patients. A total of 28 deaths (2%) occurred (Table 4) [68,69]. 

## 13. Application of CAR T Cell and CAR-NK Therapy in Solid Tumors

The CAR technology is not just limited to ALL and NHL treatment; it is very efficacious against other hematological malignancies, such as multiple myeloma (there are ongoing clinical trials of CAR-T cells in a wide array of solid tumors, including mesothelin-positive advanced refractory solid cancer, non-small-cell lung carcinoma (NSCLC), mesothelioma, gastric cancer, pancreatic cancer, esophagogastric junction cancer, hepatocellular carcinoma, glioblastoma, medulloblastoma, colon cancer, neuroblastoma, renal cell carcinoma, HER2-positive gastric cancer, HER2-positive breast cancer, bladder cancer, head and neck squamous cell carcinoma, cancer of the salivary gland, pediatric solid tumor, germ cell tumor, retinoblastoma, EGFR/B7H3-positive advanced lung cancer, EGFR/B7H3-positive advanced triple-negative breast cancer, squamous cell carcinoma of the lung, colorectal cancer, Merkel cell carcinoma, and myxoid/round cell liposarcoma. These trails have been elegantly discussed in detail [70], as well as more recent trials targeting EGFR and IL-13Ra2 in the glioblastoma [71]. 

In addition to T cells, other innate effector cells, such as natural killer cells (NK), that are used in the context of CAR (CAR-NK) have shown great anti-tumor efficacy in solid tumors [72]. For instance, CAR-NK is successful in treating acute myeloid leukemia (AML) [73]. CAR-NK in combination with other immunoregulatory treatments, such as the co-expression of IL-21 or immune checkpoint inhibitor anti-PDL-1 mAb, is effective against castration-resistant prostate carcinoma [74] and lung cancer [75], respectively. Various sources, such as residential or expanded NK (rNK and eNK) cells, as well as umbilical cord (rUC) and placental blood (rP) NK cells, were used. Residential and expanded (rNK and eNK) cells with a diverse biomarker expression profile, including the NKG2A^+^, NKG2D^+^, NKp46^+^, and NKp44^+^ subsets used in CAR technology, show efficacy against tumors [76]. 

## 14. Incorporation of Suicide Gene to Increase Efficacy of CAR Therapy

CAR-T cell therapy has been observed to cause cytokine release syndrome (CRS) or immune-effector-cell-associated neurotoxicity syndrome (ICANS), resulting in the release of potentially life-threatening toxicities. These side effects are the result of excessive immune reactions caused by the therapy [77]. Cytokine release is a mechanism that CAR-T cell therapy uses to destroy cancer cells; however, if the number of active CAR-T cells increases, the cytokines that are released can form a positive feedback loop that further increases cytokine release, resulting in a cytokine storm that causes intense body inflammation [78]. The persistence of these high cytokine levels can eventually lead to neurological toxic side effects including delirium, aphasia, hallucinations, and seizure-like activity [79]. 

To prevent toxic neurological side effects, CAR-T cell therapy should incorporate a suicide gene into the engineered CAR-T cell, making it possible to eliminate unwanted or excess CAR-T cells. The suicide gene could be implemented using gene-directed enzyme prodrug therapy (GDEPT), CRISPR/Cas9, or through mAb-mediated mechanisms. 

Gene-directed enzyme prodrug therapy consists of three things: a prodrug, a gene encoding for an enzyme that will activate the prodrug, and a carrier [80]. In GDEPT, the transgene encoding the enzyme will activate the prodrug and the prodrug will create toxic products that result in cell death [81]. GDEPT can be very effective because tumor-cell-specific promoters can control gene expression. This means that the enzyme and its associated toxic reaction can be precisely limited to tumor cells. Because of the specificity of the treatment, the therapeutic index of the prodrugs is significantly higher than that of common chemotherapy drugs [82]. This allows for the administration of higher dosages of prodrugs which, alongside CAR-T cells, could lead to increasingly durable clinical responses with decreased off-target cytotoxicity. However, special care must be taken in the selection of the encoded enzyme and the prodrug that is administered. One of the most commonly used models for GDEPT is the herpes simplex virus thymidine kinase/ganciclovir system (HSV-TK/GCV system), in which the HSV-TK gene in tumor cells, followed by a treatment with the prodrug GCV, causes tumor cell apoptosis [83]. The system is successful; however, because HSV-TK has a viral origin, it could activate the immune system, causing the eventual elimination of the CAR-T cells [84]. 

Clustered, regularly interspaced, short palindromic repeats (CRISPR)/Cas9 also showed promise in increasing the efficacy of CAR-T cell therapy. CRISPR/Cas9 is a gene-editing tool that allows sequence-specific changes to be made to human DNA. This technology has been shown to improve CAR-T cell therapy by directing a CD19 CAR to the T-cell receptor alpha constant (TRAC) locus. The additional manipulation of CAR by CRISPR/Cas9 resulted in CAR-T cells that outperformed conventionally engineered CAR-T cells in a mouse model with acute lymphoblastic leukemia [85]. The CRISPR/Cas9-edited CAR-T cells showed more uniform expression in human peripheral blood T cells, increased T-cell potency, and mechanisms that delayed effector T cell differentiation and exhaustion [80]. Moreover, this technology could allow for the insertion of a suicide gene into CAR-T cells, which could be recognized by an enzyme that would be administered following treatment completion. This could eliminate all residual active CAR-T cells, thereby preventing excess cytokine release after the treatment ends. 

Therapeutic mAb-mediated mechanisms can also be added to traditional CAR-T cell therapy to further increase its efficacy. The CD20 molecule is proposed to have the ability to function as a suicide gene for T lymphocytes due to its dual function as a selection marker as well as a killer gene following its exposure to rituximab (the anti-CD20 therapeutic antibody [81]). One study transduced CD20-positive T lymphocytes from wild-type human CD20 cDNA using a Moloney-derived retroviral vector. A total of 86–97% of the transformed CD20-positive T lymphocytes were killed in vitro following the administration of rituximab and its complement. Moreover, the addition of the CD20 transgene did not alter the function of the T lymphocytes [86]. This finding suggests that if CAR-T cells could be manipulated to express CD20 receptors, the CAR-T cells that remained in circulation after treatment completion could be easily eliminated with the administration of rituximab, without fear of decreasing the functional efficacy of the CAR-T cells (Figure 3). 

## 15. FDA-Approved Small-Molecule Inhibitors to Increase the Efficacy of CAR T Cell Therapy

### 15.1. Celecoxib (Celebrex, COX2 Inhibitor)

Inflammatory responses are initiated through the conversion of arachidonic acid to prostaglandin via cyclooxygenase enzymes (COX1 and COX2). In the cell membrane, prostaglandin is further converted to prostacyclin (PGI2) to cause vasodilation and inhibit platelet aggregation, as well as Thromboxane A2 (TXA2), which is used to cause vasoconstriction and promote platelet aggregation. Upon the completion of these reactions, an inflammatory response is initiated. There are several ways to compete and fight exogenously with this mechanism and one of them is to inhibit COX2. Theoretically, there are structural and functional differences between COX 1 and 2. COX1 works as a housekeeping gene and is expressed in almost all tissue, where it produces prostaglandins to control homeostatic functionalities in the tissues. On the other hand, COX2, which is encoded by the Ptgs2 gene, reaches its maximum induction during inflammation in cells experiencing inflammatory arthritis, pro-inflammatory cytokines, and tumorigenic potential [87]. COX2 enzyme inhibitors have been in the market for decades and the most efficient and FDA-approved version of them is Celecoxib (a nonsteroidal anti-inflammatory drug-NSAID), which is the preferred drug of interest when it comes to COX2 inhibitors. Other drugs, such as Diclofenac and Meloxicam, have the same mechanism of action as Celecoxib, but celecoxib is the preferred drug of interest approved by the FDA for the treatment of osteoarthritis, rheumatoid arthritis, ankylosing spondylitis, and primary dysmenorrhea [84,85,86,87] (Figure 3). 

### 15.2. Histone Deacetylase Inhibitors (HDACi) Vorinostat and Panobinostat

Vorinostat (SAHA) and Panobinostat are histone deacetylases that belong to hydroxymates. Eukaryotic cells have characteristic processes, such as histone acetylation and deacetylation, that play major roles in the regulation of transcription. To maintain a balance between these two processes to promote normal cell growth, two distinct enzymes, called histone acetyltransferase and deascetylase, participate in the further development of different diseases such as cancer [79]. HDACi has several effects, both in vivo and in vitro, specifically arresting growth, affecting cell differentiation, and causing the apoptosis of malignant cells. Moreover, both monotherapy and a combination therapy of these drugs with anti-neoplastic drugs have achieved positive outcomes in curing cutaneous T-cell lymphomas (CTCL). However, the mechanisms of action of HDACi are broad; they inhibit class I and II HDAC enzymes but not class III [80]. Crystallographic studies show that Vorinostat binds to the zinc atom of the catalytic site of the HDAC enzyme while the phenyl ring of Vorinostat projects out of the catalytic domain onto the surface of the HDAC enzyme [79]. Furthermore, this binding causes an accumulation of acetylated histones and exhibits multiple effects on transcriptional and non-transcriptional processes. Moreover, HDACi can regulate apoptotic genes, allowing for them to be effective anti-tumor agents. HADCs regulate apoptosis via both extrinsic and intrinsic pathways. The extrinsic pathway of HDACi operates via the upregulation of death receptor 5 (DR5), and the death-inducing receptor of tumor-necrosis-factor-related apoptosis-inducing ligand (TRAIL) [81]. TRAIL induces apoptosis in tumor cells and has the luxury of leaving the untransformed cells mostly unaffected. HDACs also regulate the intrinsic apoptotic pathway via a reduction in the expression of key anti-apoptotic proteins such as Bc1-_xL_, Mc1 and XIAP, as well as an increase in the expression of pro-apoptotic proteins such as Bim, Bax, and Bak [82] (Figure 3). 

## 16. Regulation of Apoptotic Machinery by Cox-2 Inhibitor and HDACi

### 16.1. Celecoxib

Celecoxib-induced apoptosis is mediated via the intrinsic apoptotic signaling pathway, which is Bcl-2 independent but apoptosome-dependent. Apoptosome is a large quaternary protein structure that is formed in the process of apoptosis upon cytochrome c release from the mitochondria in response to either intrinsic or extrinsic cell death stimuli. Celecoxib-induced apoptosis requires Apaf-1 and pro-caspase-9 in Jurkat T lymphoma cells, but the absence of Bcl-2 has no effect on apoptosis induced by celecoxib. Moreover, the unaltered size and quantity of the non-phosphorylated Bcl-2 protein levels show that the overexpression of Bcl-2 does not affect celecoxib-induced apoptosi. Celecoxib interferes with pro-survival signals by downregulating Mci-1. Also, a sharp decline in Mcl-1 levels in celecoxib-treated Jurkat was observed, allowing for Bak to trigger apoptosis [83]. Indeed, the increased Bcl-_xL_ levels respond to the functionality of Mcl-1, which blocked apoptosis in Mcl-1-deficient cells. On the other hand, the presence of Bak is more important than the presence of Bcl-2 for celecoxib to trigger apoptosis. In conclusion, the intrinsic signaling pathway is the dominant mechanism that enables celecoxib to induce apoptosis in the presence of functional Bak [84]. These results suggest that HDACi can be effectively used as an adjuvant (immunosesitizing agent) in CAR-T cell settings [84,85,86,87] (Figure 4).

### 16.2. Vorinostat (SAHA)

After Vorinostat binds to the HDAC enzyme, acetylated histone protein that exhibit several cellular effects at both transcriptional and non-transcriptional levels accumulate. The transcriptional effect occurs via the direct binding of Vorinostat to the HDAC enzyme, which acts on several transcription factors, such as E2F-1, YY-1, Smad7, p53, Bcl-6, and GATA-1 [88]. This binding alters the expression of certain genes, such as the acetylation of Bcl-6 transcriptional activator, which initiates the inhibition of transcriptional repression by Bcl-6 [89]. On the other hand, non-transcriptional effects include cell cycle arrest, apoptosis, the inhibition of angiogenesis, which is the formation of blood vessels, and, finally, the downregulation of immunosuppressive interleukins. 

As mentioned above, vorinostat causes apoptosis in hematological malignancies and tumors via both transcription and transcription-independent pathways. HDAC inhibition causes an imbalance in the ratio of pro- and anti-apoptotic proteins that allow for the apoptosis signal to proceed [89]. HDACi upregulates DR5 and sensitizes tumors to TRAIL-induced apoptosis in TRAIL-resistant cells. Moreover, Vorinostat downregulates proteins like anti-apoptotic proteins, which are responsible for maintaining mitochondrial integrity, thus assisting in apoptosome formation. It also upregulates proteins like Bim, Bak, and Bax, facilitating the full execution of the apoptotic signals. Finally, hyperacylation has a stabilizing effect on p53, further regulating apoptosis in CTCL [90]. Therefore, through their broad gene-regulatory effects, HDACi can regulate apoptosis in tumor cells and can be effectively used as an adjuvant in clinical settings [90,91,92,93] (Figure 4). 

## 17. Conclusions/Future Directions

Traditional treatment protocols for NHL and ALL have been significantly revised and improved over time, resulting in higher response rates. However, these modalities are limited, mainly due to the undesired severe toxicity of the treatment and the presence of inherent (primary) or development of acquired (secondary) tumor resistance to conventional therapies. This spurred the development of CD19-redirected CAR T cells with potent anti-tumor activity. Clinical trials of various FDA-approved CD19CAR T cell therapies, namely, tisagenlecleucel, Brexucabtagene autoleucel KTE-X19 (Tecartus), Lisocabtagene Maraleucel (Breyanzi), and axicabtagene ciloleucel, conducted at various institutions, have achieved significant success in treating NHL and ALL patients. Despite their initial success, these responses are short-lived due to the primary or secondary resistance of malignant B cells, as well as the toxic, treatment-associated side effects. 

Several approaches are being considered to increase the efficacy of CD19CAR T cell therapy. One general approach to improvement could be optimizing the number of infused, transgenic CD19 CAR T cells. Another approach is the incorporation of FDA-approved small molecules with broad gene-regulatory effects in conjunction with CD19CAR T cells. Epigenetic modifiers such as HDACi and the anti-inflammatory drug Celecoxib can regulate the apoptotic machinery and create a proapoptotic tumor milieu, thus reducing the apoptosis threshold of resistant tumor cells, which, in turn, will allow for the successful execution of apoptotic death signals delivered by transgenic T cells. This approach may alleviate the need for a higher number of transgenic T cells to be infused into the patient, thus reducing the side effects. The persistence of infused CD19CAR T cells after the cessation of treatment is another major concern in patients receiving CD19CAR T cell therapy. Engineered T cells will cause a multitude of side effects, including hallucinations, seizure-like activities, fever, delirium, aphasia, hypotension, and hypoxia. The incorporation of a suicide gene into the genetically engineer CD19CAR redirected T cells using GDEP-, CRISPR/iCasp9-, or mAb-mediated mechanisms will assist in curbing the undesired long-term side effects. Future research should investigate these approaches to further improve CD19CAR T cell efficacy in the treatment of hematological malignancies of B cell origin. 

## Figures and Tables

**Figure 1 cells-13-00662-f001:**
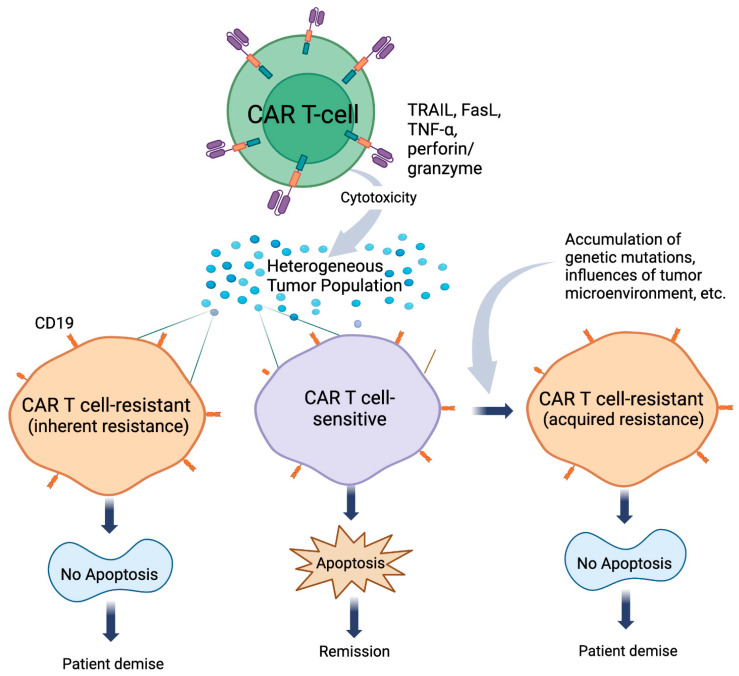
CAR-T-cell-mediated killing of tumors. Tumor cells are composed of heterogenous cell populations. CAR-T cells redirected against specific surface markers (e.g., CD19 in NHL) will induce apoptosis in a subpopulation of tumors (sensitive tumor cells) through Fas/FasL, TRAIL death receptors/TRAIL, TNF-R1 and -R2/TNF-α, and perforin granzyme pathways. This will result in patients undergoing remission. Alternatively, a subpopulation of the original tumor cells is either inherently resistant or develops resistance (acquired resistance) to the apoptotic stimuli delivered by CAR-T cells. Acquired and inherent CAR-T-cell-resistant tumors will lead to patient demise.

**Figure 2 cells-13-00662-f002:**
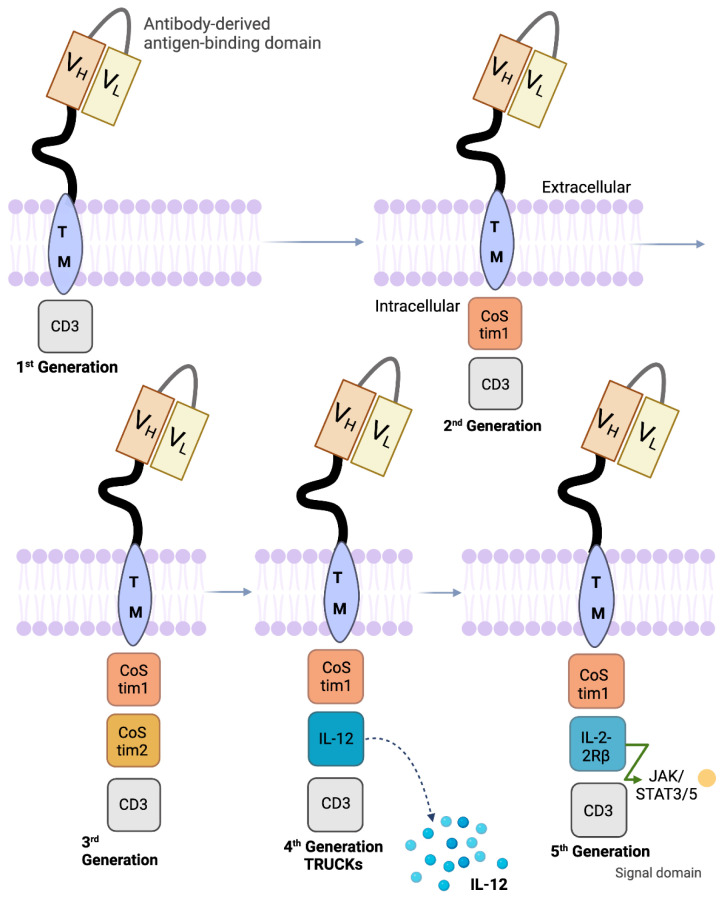
Different generations of chimeric antigen receptor (CAR) constructs. The structure and composition of various CAR constructs (first-, second-, third-, fourth- [TRUNK] and fifth-generation CAR constructs are depicted. Refer to the text for detailed information. VH: variable region of heavy chain; VL: variable region of light chain of mAb.

**Figure 3 cells-13-00662-f003:**
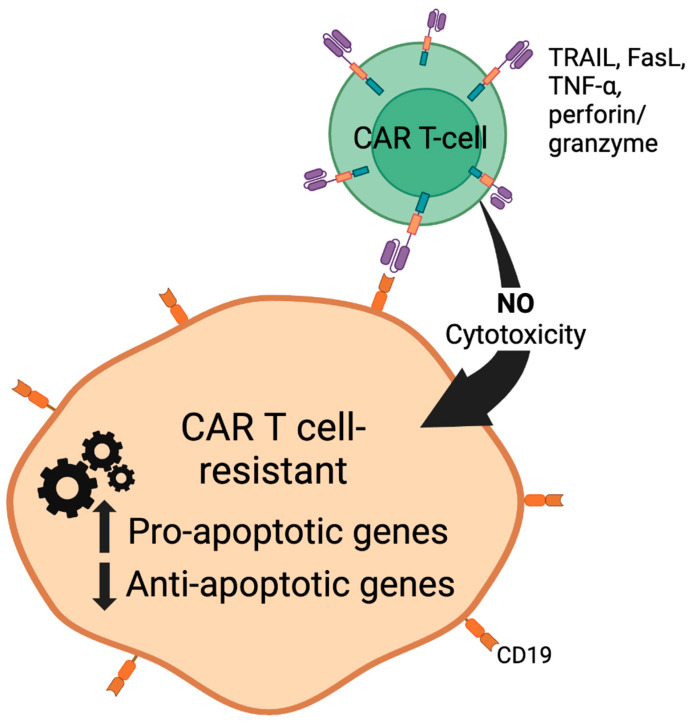
Development of CAR-T cell resistance. A subpopulation of tumor cells is inherently (innately) resistant to CAR-T-cell-induced apoptosis. Alternatively, prolonged CAR-T cell exposure will result in the selective expansion of tumor cells that developed resistance to CAR-T cells (acquired resistance) after successful initial CAR-T cell therapy. These tumors exhibit an altered expression profile of pro- and anti-apoptotic gene products.

**Figure 4 cells-13-00662-f004:**
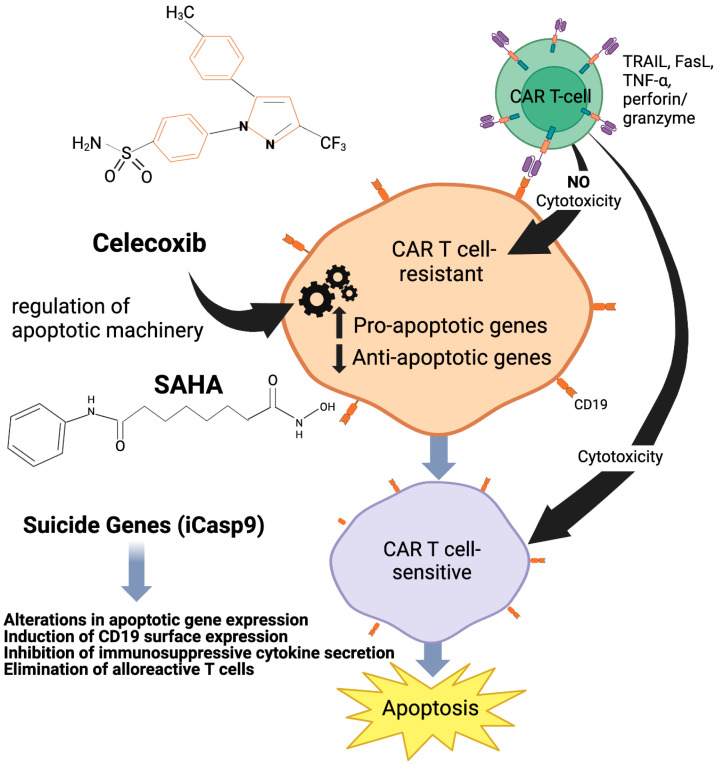
Proposed model of the mechanism of Celecoxib and HDACi-mediated sensitization to CAR-T-cell-induced apoptosis. Both Celecoxib (Celebrex) and HDACi (SAHA, LBH589) exhibit broad apoptosis gene-regulatory effects. The pretreatment of CAR-T-cell-resistant tumors with subtoxic concentrations of these FDA-approved agents can modify the dynamics of cell survival signaling pathways and alter the expression profile of apoptotic genes. By favoring a proapoptotic tumor milieu, tumors will become sensitive to the apoptotic effects of these agents. Also, the incorporation of suicide genes (iCasp9) can render tumors sensitive to CAR-T cell-killing. Refer to the text for more detailed information.

**Table 1 cells-13-00662-t001:** Summary of CD19 CAR-T cell clinical trials in ALL and NHL patients.

Summary of Trial	Drug/ Treatment	Overall Response Rate (ORR)	Objective Response Rate (ORR)	Complete Remission (CR)	Disease Free Progression (DFP)	Toxicites (Most Common)	References
*n* = 30 Children/ Adults r/r ALL	CTL019 (Kymriah)	OS = 78%		90%	Sustained remissions at 6-month event -free survival = 67% Predicted probability of persistence with CTL019 = 68% Probability of relapse-free B-cell aplasia = 73%	Severe: CRS = 27% Any grade: CRS = 100% NT = 43% B-cell aplasia In patients with a response = 100%	[61]
*n* = 28 DLBCL or r/r FL	CTL019 (Kymriah)	64%		FL = 71% DLBCL = 43%	Sustained remissions of patients with response at 28.6 month (median) follow-up: FL = 89% DLBCL = 86%	Severe: CRS = 18% NT = 11% Any grade: CRS = 57% NT = 39%	[62]
*n* = 93 adults r/r DLBCL (JULIET)	Tisagenlecleucel (Kymriah)	52%		Complete response 40%	12-month post first response: Relapse-free survival = 65% CR relapse-free survival = 79%	Severe: CRS = 22% Neurologic events = 12% Cytopenias (>28 days) = 32% Any grade: CRS = 58% Neurologic events = 21% Cytopenias = 44%	[58]
*n* = 101 r/r DLBCL, PMBCL or (transformed) FL (ZUMA-II)	Axicabtagene Ciloleucel (Yescarta)		82%	54%	Sustained remissions with a 15.4-month (median) event-free survival = 42% CR at 15.4 month (median) = 40% OS at 18-months = 52%	Severe: Thrombocytopenia = 38% Neutropenia = 78% Anemia = 43% CRS = 13% Neurologic events = 28%	[59,60]
*n* = 20 adults r/r CLL or ALL	CAR T-19/ CTL019 (Kymriah)	CLL = 42.9% ALL = 83.3%		CLL = 21.4% ALL = 83.4%		Severe: CRS (CLL) = 42.86% CRS (ALL) = 83.33%	[56]
*n* = 7 Refractory DLBCL	KTE-C19 (Yescarta)	71%	N/A	57%	CR at 12+ months 0.43	Severe: CRS = 14% NT = 57% Any grade: CRS = 85% NT = 44%	[57]
*n* = 53 adults Relapsed B-cell ALL	Autologous 19 − 28z + CAR T-cells	N/A	N/A	83%	29-month (median) Follow-up: Event-free survival = 6.1 months (median) OS = 12.9 months (median)	Severe: CRS = 26% NT = 42% Any grade: CRS = 85% NT = 44%	[63]
*n* = 59 Children/adults r/r ALL	CTL019 (Kymriah)	OS at 12 month (median) Follow-up = 79%	93% (1-month Post Infusion)	N/P	12-month (median) Follow-up: CR = 58%	Severe: CRS = 27% B-cell aplasia (continuing CR patients) = 71% Any grade: CRS = 88%	[64]
*n* = 75 Pediatric/young Adult r/r B-cell ALL (ELIANA)	Tisagenlecleucel (Kymriah)	81% (3 mo. ≥)	PR = 21%	3-month overall remission rate = 81% CR = 60% Incomplete hematologic recovery + CR = 21%	6-month follow-up: Patients with relapse-free survival = 80% OS = 90% 12-month follow-up: patients with relapse-free survival = 59% OS = 76%	B-cell aplasia in Patients with a response = 100% Severe: CRS = 46% Neurologic event = 13% Infection = 24% Any grade: CRS = 77% Neurologic event = 40% Infection = 43%	[57]

ALL: acute lymphocytic leukemia; CLL: chronic lymphocytic leukemia; CR: complete response; CRS: cytokine release syndrome; DLBCL: diffuse large B-cell lymphoma; FL: follicular lymphoma; NT: neurotoxicity; OS: overall survival; PMBCL: primary mediastinal B-cell lymphoma; r/r: relapse/refractory; SD: stable disease.

**Table 2 cells-13-00662-t002:** A. Summary of clinical trials of tisagenlecleucel in the treatment of NHL. B. Summary of clinical trials of tisagenlecleucel in the treatment of ALL.

Summary of Trial	Infusion Amount (Per kg of Body Weight)	ORR%	CR%	PR%	DFP%	Toxicities ≥ 3	Mortalities Post Infusion	Reference
*n* = 75 (ELIANA)	1.0 × 108 (median)	81% (3 mo. ≥)	60	21	59 (12 mo.)	CRS: 46 NT:13 FN:35 TLS:4	3 (CRS and other AE), 13 (DP + subsequent therapies, etc), 1 (unknown)	[57]
*n* = 30	0.76 × 106 To 20.6 × 106	78	90	NP	67 (8 mo)	CRS: 27	7 (DP)	[61]
*n* = 59	1 × 107 to 1 × 108	79 (12 mo.)	NP	93 (at 1 mo.)	58 (12 mo.)	CRS: 27	NP	[64]
*n* = 63	>50 kg infusion amt = 0.1 − 2.5 × 108 or ≤50 kg infusion amt = 0.2 − 5 × 106 (single doses)	83	63	NP	Not reached at 4.8 mo. (median) follow-up	CRS: 49 NT: 18 (safety population *n* = 68)	0	[54]
*n* = 6 (ELIANA subgroup)	>50 kg infusion amt = 0.1 − 2.5 × 108 or ≤50 kg infusion amt = 0.2 − 5.0 × 106 (single doses)	66.7 (3 mo.)	42.8	NP	100 (median time of 6 mo.)	CRS: 5 NT: 0	2 (DP, 71, and 352 days post infusion)	[65]
**Summary of** **trial**	**Infusion** **amount (per kg** **of body weight)**	**ORR%**	**CR%**	**PR%**	**DFP%**	**Toxicities** **≥3**	**Mortalities post** **infusion**	**Reference**
*n* = 93 r/r DLBCL, tFL or HGBL-DH/TH (JULIET)	3.0 × 108 (median)	52	40	12	65 (12 mo.)	CRS:22 NT:12 FN:15 TLS:1	3 (DP, 30 days post infusion)	[58]
*n* = 28 DBCL or r/r FL	3.1 × 106 to 8.9 × 106	64	FL: 71 DLBCL: 43 Combined: 57	18 (at 3 mo.)	FL:89 DLBCL: 86 (28.6 mo)	CRS:18 NT: 11	1 (NT)	[62]
*n* = 9 r/r DLBCL (JULIET subgroup)	2.0 × 108 (median)	77.8	55.6	22.2	100 (25 − 550 + day range)	CRS: 2 NT: 1	2 (DP, 30 days post infusion	[66]

**Table 3 cells-13-00662-t003:** Summary of Clinical Trials of Tisagenlecleucel and Axicabtagene Ciloleucel for the Treatment of Refractory NHL and ALL.

Summary of Trial	Infusion Amount (Per kg of Body Weight)	ORR%	CR%	PR%	DFP%	Toxicities ≥3	Mortalities Post Infusion	Reference
*n* = 108 r/r DLBCL, tFL or PMBCL (ZUMA-I & II)	2.0 × 106	82 (12 mo.)	58	29	41 (15 mo.)	CRS: 13 NT:28	2 (CRS), 42 (DP + subsequent therapies, etc.)	[59,60]
*n* = 53 r/r DLBCL or FL	1 × 106 or 3 × 106	-	83	-	-	CRS: 26 NT:42	1	[63]
*n* = 22 r/r DLBCL, FL, or mantle cell lymphoma	1 × 106, 2 × 106 or 6 × 106	73	55	18	63.3 (12 mo.)	NT: 55	0	[67]

AE; adverse effects, CR; complete response, CRS; cytokine release syndrome, DFLBCL; diffuse large B-cell lymphoma, DFP; disease-free progression, DP; disease progression, FN; febrile neutropenia, HGBL- DH/TH; high-grade B-cell lymphoma with MYC rearrangement, plus rearrangement of BCL2, BCL6, or both genes, NP; not provided, NT; neurologic toxicities, ORR; overall response rate, PR; partial response, PMBCL; primary mediastinal B-cell lymphoma, TC; thrombocytopenia, tFL; transformed follicular lymphoma, TLS; tumor lysis syndrome.

**Table 4 cells-13-00662-t004:** A. Summary of brexucabtagene autoleucel CD19 CAR-T cell clinical trials in ALL and NHL patients. B. Summary of clinical trials of Tecartus in the treatment of ALL. C. Summary of clinical trials of Tecartus in the treatment of NHL.

Summary of Trials		Drug/ Treatment	Overall Response	Objective Response	Complete Remission	Disease Free Progression	Toxicities (Most Common)	Reference
*n* = 54 *n* = 78 r/r B-ALL (all leukapheresed patients) r/r B-cell precursor (ALL) (ZUMA 3, Phase II)		Brexucabtagene autoleucel KTE-X19 (Tecartus)	83.60% (16.4% of people showed no response)	71%	52%	Median duration of remission (DOR) = 14.6 mo. Median relapse-free survival (RFS) = 11.6 mo. Median overall survival (OS) = 25.4 mo	Total = 79% Severe: CRS = 26% NT = 35% Any grade: CRS = 92% NT = 87%	[62,65]
*n* = 68 r/r mantle cell lymphoma (NHL) (ZUMA 2, Phase II)		Brexucabtagene autoleucel KTE-X19 (Tecartus)	91%	91%	68%	Median OS = 46.6 mo.	Grade 1 or 2: CRS = 76% NT = 32% Severe: CRS = 15% NT = 31%	[66,67]
*n* = 92 LBCL (NHL) (TRANSFORM, Phase III)		Lisocabtagene maraleucel (Breyanzi)	87%		74%	Not reached (NR)	Grade 3: CRS: 1% NT: 4% No grade 4 or 5 events Prolonged cytopenia = 43%	[63]
**Summary**	**Infusion** **amount (per kg** **of body weight)**	**ORR%**	**CR%**	**PR%**	**DFP%**	**Toxicities** **≥3**	**Mortalities post** **infusion**	**Reference**
*n* = 54 *n* = 78 r/r B-ALL (ZUMA 3, Phase II)	1 × 10^6^	83.6	52	15	39 (complete remission)	97% Febrile neutropenia: 35% Infections: 30% CRS: 26% NT: 35%	Fatal adverse reactions = 5% (cerebral edema and infections)	[62,65]
**Summary**	**Infusion** **amount (per kg** **of body weight)**	**ORR%**	**CR%**	**PR%**	**DFP%**	**Toxicities** **≥3**	**Mortalities post** **infusion**	**Reference**
*n* = 68 r/r mantle cell lymphoma (NHL) (ZUMA 2, Phase II)	2 × 10^6^	91	68	24	24.9— to not estimable	99% Cytopenias: 94% CRS: 91% NT: 63%	No death from CRS No death from NT 16 deaths total (24%): Death from progressive disease = 14 patients (21%) Grade 5 AE = 2 patients (3%)	[66,67]

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
