# Peer review of "Immunotherapy of Hematological Malignancies of Human B-Cell Origin with CD19 CAR T Lymphocytes"

_cells, 2024, doi:10.3390/cells13080662_

Round 1

Reviewer 1 Report

Comments and Suggestions for Authors

Present study highlights the overall overview of the haematological malignancy related to Acute lymphoblastic leukemia (ALL) and non-Hodgkin's lymphoma (NHL) and their possible outcome after the therapies. Here authors points about the most recent and effective CAR-T cells and possible application of the blood cancer. One of the positive aspects of this study was the author describe the process very well and descriptive.  Below are some comments to clarify and make this work in better presentable format.

Comments

-       It will be good for the readers if the author can make some pictorial representation of the study. e.g.; in section-5: design of various CD19 CAR constructs- please make the picture through BioRender for better representation etc. 

-       Rearrange the table in proper size and the text inside it, it is difficult to corelate the information what is presented in this format. 

Author Response

We very much appreciate the constructive suggestions raised by reviewer 1.

  As suggested, using BioRender software, we have created 4 figures and have added them to the newly submitted revised manuscript. Additionally, we have reformatted the tables. The editorial staff have also assured us to further reformat the tables if needed.   

Once again, We are very grateful for the constructive suggestions; responding to these critiques have made the manuscript more comprehensive. 

Reviewer 2 Report

Comments and Suggestions for Authors

The authors prepared a well-thought review that describes the current status of Acute Lymphoblastic Lymphoma (ALL) and Non-Hodgkin’s Lymphoma (NHL). The manuscript can be divided into two main sections. In the first section, the authors describe the two malignancies and standard treatment plans, including drug options and overall survival rate. They analyze the benefit of standard treatments, highlighting at the same time the issue related to relapsed or refractory cases. The second section introduces the chimeric antigen receptor (CAR) T-cell immunotherapies and describes their application in ALL and NHL. Data from clinical trials add a significant value to the review. Overall, the manuscript is well-organized and written, insightful, easy to follow, and provides key information on CD19 CAR T-cell therapy. Before publication, I would highly recommend reorganization of the tables, as these are poorly prepared. An illustration presenting the key features of CAR constructs (see section 5), differences across the multiple generations, and an overall mechanism of action would be very helpful to readers. Discussion on the fifth generation of CAR constructs should be included as this will increase the quality of the manuscript.

Author Response

We very much appreciate the constructive suggestions raised by reviewer 2.

As suggested, we have created 4 figures and have added them to the newly submitted revised manuscript. These figures illustrate:

 Figure 1: CAR-T cell-mediated killing of tumors.

Figure 2: Different Generations of Chimeric Antigen Receptor (CAR) Constructs. 

Figure 3: Development of CAR-T cell resistance

Figure 4: Proposed Model of Mechanism of Celecoxib and HDACi-mediated sensitization to CAR-T cell induced apoptosis.

Additionally, we have provided information about fifth generation of CARs and have added reference 51 to support the statements as well as references 71-77 to provide more detailed information on CAR therapy in other types of cancers. Furthermore, we have reformatted the tables. The editorial staff have also assured us to further reformat the tables if needed.   

Once again, We are very grateful for the constructive suggestions; responding to these critiques have made the manuscript more comprehensive. 

Reviewer 3 Report

Comments and Suggestions for Authors

The authors of the Review “Immunotherapy of Hematological Malignancies of Human 2 B-Cell Origin with CD19 CAR T Lymphocytes” describe the use of genetically engineered chimeric antigen receptor (CAR) T-cells to treat patients acute lymphoblastic leukaemia and non-Hodgkin's lymphoma. 

The manuscript is well structured. Authors were able to highlight current information on conventional treatments as well as to describe the novelties in the designing and manufacturing of various generations of CAR T-cells. 

The data are comprehensive and all tables reported to summarize the clinical trials are clear and complete.

Overall the review is very well written.

Minor criticisms

1.     There are a few typos throught the text 

2.     It would be useful to add a figure at the beginning of the paper to better explain the role and the differences amongst CAR T-cells 

Author Response

We very much appreciate the constructive suggestions raised by reviewer 3.

As suggested, we have included 4 figures and have added them to the newly submitted revised manuscript. These figures illustrate:

 Figure 1: CAR-T cell-mediated killing of tumors.

Figure 2: Different Generations of Chimeric Antigen Receptor (CAR) Constructs. 

Figure 3: Development of CAR-T cell resistance

Figure 4: Proposed Model of Mechanism of Celecoxib and HDACi-mediated sensitization to CAR-T cell induced apoptosis.

We have also edited the manuscript for typographical errors.

Once again, We are very grateful for the constructive suggestions; responding to these critiques have made the manuscript more comprehensive.